# Ketogenic Diet and Ketone Bodies as Clinical Support for the Treatment of SARS-CoV-2—Review of the Evidence

**DOI:** 10.3390/v15061262

**Published:** 2023-05-27

**Authors:** Izabela Bolesławska, Magdalena Kowalówka, Natasza Bolesławska-Król, Juliusz Przysławski

**Affiliations:** 1Department of Bromatology, Poznan University of Medical Sciences, 60-806 Poznan, Poland; mkowalowka@ump.edu.pl (M.K.); jprzysla@ump.edu.pl (J.P.); 2Student Society of Radiotherapy, Collegium Medicum, University of Zielona Gora, Zyta 28, 65-046 Zielona Góra, Poland; natmeff@gmail.com

**Keywords:** COVID-19, ketogenic diet, ketone bodies, SARS-CoV-2

## Abstract

One of the proposed nutritional therapies to support drug therapy in COVID-19 is the use of a ketogenic diet (KD) or ketone bodies. In this review, we summarized the evidence from tissue, animal, and human models and looked at the mechanisms of action of KD/ketone bodies against COVID-19. KD/ketone bodies were shown to be effective at the stage of virus entry into the host cell. The use of β-hydroxybutyrate (BHB), by preventing the metabolic reprogramming associated with COVID-19 infection and improving mitochondrial function, reduced glycolysis in CD4+ lymphocytes and improved respiratory chain function, and could provide an alternative carbon source for oxidative phosphorylation (OXPHOS). Through multiple mechanisms, the use of KD/ketone bodies supported the host immune response. In animal models, KD resulted in protection against weight loss and hypoxemia, faster recovery, reduced lung injury, and resulted in better survival of young mice. In humans, KD increased survival, reduced the need for hospitalization for COVID-19, and showed a protective role against metabolic abnormalities after COVID-19. It appears that the use of KD and ketone bodies may be considered as a clinical nutritional intervention to assist in the treatment of COVID-19, despite the fact that numerous studies indicate that SARS-CoV-2 infection alone may induce ketoacidosis. However, the use of such an intervention requires strong scientific validation.

## 1. Introduction

Viral infections are a constant threat to public health. Morbidity and mortality associated with seasonal influenza A epidemics but also regularly emerging new viruses with pandemic potential such as Severe Acute Respiratory Syndrome (SARS), N5N1 influenza, H1N1 influenza, Middle East Respiratory Syndrome (MERS) and, most recently, Severe Acute Respiratory Syndrome Coronavirus 2 (SARS-CoV-2) [1] is a key global healthcare problem, requiring the identification of new therapies capable of reducing the severity of these infections.

The SARS-CoV-2 pandemic, which has caused a wave of illness and deaths worldwide as of 2019 [2,3], has not yet been fully contained [4,5]. Although concentrated efforts by researchers and clinicians have led to the identification of new therapeutic options [4,6] and vaccines [5], full immunization with vaccines does not appear to be sufficiently effective against new variants of SARS-CoV-2 [7,8], and treatment is still mainly based on symptomatic treatment [9,10]. This has prompted the search for innovative solutions that could prevent virus entry, strengthen the body’s protective barrier, support COVID-19 treatment and recovery from the disease, and protect against new SARS-CoV-2 strains.

Numerous studies conducted in this area have pointed to the prophylactic role of adequate nutrition in reducing the risk of SARS-CoV-2 infection [11,12,13]. More recently, the role of clinical nutritional therapy as support for drug therapy in improving the course of COVID-19 has also been highlighted [14].

One of the dietary options being considered is the ketogenic diet (KD), which aims to induce physiological ketosis [1] by significantly restricting the offset of carbohydrates by high-fat content. Ketone bodies resulting from KD, such as β-hydroxybutyrate (BHB), acetoacetate (AcAc), and acetone, affect the body by producing, for example, anticonvulsant effects [15,16] used in the treatment of drug-resistant epilepsy since the early 20th century [17,18,19]. There are also an increasing number of studies confirming the neuroprotective effects of KD/ketone bodies [20,21,22], their efficacy in the prevention of Parkinson’s and Alzheimer’s disease [23,24], the treatment of obesity [25,26,27,28], type 2 diabetes [26,28], cardiovascular disease [29,30,31], or cancer [28,32,33]. Several studies have also confirmed the immunostimulatory and anti-inflammatory effects [14,34,35,36,37,38,39] of KD and ketone bodies. Induction of a state of ketosis by KD or ketone bodies and their pleiotropic effects may also prove to be an effective intervention in the prevention and/or treatment of COVID-19 coronavirus disease [40,41,42,43,44,45].

KD and beta-hydroxybutyrate (BHB), accounting for approximately 70% of the circulating pool of ketone bodies [46], exhibit a variety of molecular effects including metabolic regulation and effects on modulation of the epigenome, resulting in changes in DNA methylation, chromatin histone modifications and miRNA levels [46,47], and enhancement of cellular resistance to oxidative stress through multiple mechanisms [46,48] including regulation of the miRNA expression profile [49]. However, the molecular mechanisms underlying the effects of these diets are only partially understood and can result in both physiologically beneficial outcomes and potentially harmful effects [46,47]. Of particular concern is the potential for diabetic ketoacidosis leading to coma and even death (with ketone body concentrations exceeding 20 mM) [46,50]. There is a bidirectional relationship between SARS-CoV-2 and diabetes—SARS-CoV-2 can directly damage pancreatic beta cells, exacerbating hyperglycemia, causing or exacerbating diabetes in a significant number of patients, while the presence of diabetes can exaggerate a COVID-19 infection [51,52,53]. In such patients, the use of KD could exacerbate diabetes and worsen COVID-19.

Given the specificity of SARS-CoV-2, hard evidence is needed to support the actual efficacy of KD and/or ketone bodies against the virus. A review of the evidence and a look at the proposed mechanisms of action of KD/ketone bodies against SARS-CoV-2 virus was the aim of this manuscript. Notably, since the SARS-CoV-2 virus shares similarities with other viruses, including those from distant viral families [54], it causes co-infections with other viral diseases leading to worsening clinical outcomes of COVID-19 [55]. There are also concerns that the SARS-CoV-2 virus, despite pandemic containment by vaccination, could become a persistent threat to global health like the Middle East Respiratory Syndrome (MERS) coronavirus before it [56] or the Zika virus [57]. Whether in the form of an entirely new variant or evolving from the Omicron sublineage with new antigenic changes, the new variants will exhibit variability in clinically relevant features, including sensitivity to drug therapies [58]. A dietary approach using KD or ketone bodies that can change the metabolic environment to one that is unfavorable to the entry and existence of the virus in the human body may prove effective in both prevention and treatment support.

## 2. Course of SARS-CoV-2 Infection

SARS-CoV-2 (Severe Acute Respiratory Syndrome Coronavirus 2) is a single-stranded enveloped virus with positive ssRNA polarity [59,60,61,62]. Compared to previously known coronaviruses, SARS-CoV-2 has high virulence and a higher transmission rate [62].

After contact with the cell, SARS-CoV-2 virus can enter the cell either through endosomes or through plasma membrane fusion. In both cases, spike proteins (S1/S2) present on the surface of SARS-CoV-2 mediate attachment to the cell membrane by binding to angiotensin-converting enzyme type 2 (ACE2) as an entry receptor [63,64]. This is made possible by the characteristic structure of the structural spike protein (S) that forms the outer envelope of the virus, consisting of three subunits: S1, S2, and S2’. The S1 subunit via the Receptor Binding Domain (RBD) of the virus binds to the ACE2 receptor on the cell surface [65,66,67] and then host cell surface proteases such as TMPRSS2 (trans-membrane serine protease 2), cathepsin, furin, or yet other proteins (e.g., CD147, GRP78, ADAM17) act on critical cutting sites at the S1/S2 interface of the spike protein and at the S2’ site [61,65,68,69,70,71]. This results in membrane fusion and viral infection [59]. The interaction between the SARS-CoV-2 coronavirus spike protein and the host cell ACE2 receptor is shown in Figure 1.

Once the virus has entered the host cell and released its genetic material, the viral proteins necessary for further replication and translation are synthesized. The formed new viral particles, capable of entering further host cells, are translocated close to the cell membrane and released exocytosis [63,73].

The course of the disease itself and its clinical picture depends primarily on the nature and type of immune response to infection. This is made up of many factors including individual genetic predisposition, specific immune response, and the inflammation present [74].

SARS-CoV-2 virus infection stimulates several immune response pathways in the host, including activation of both innate and acquired immune responses [75]. The defender against SARS-CoV-2 is the innate immune system eliminating SARS-CoV-2 in the best-case scenario without activation of the adaptive immune system [74,76]. However, an effective immune response against SARS-CoV-2 often requires activation of both arms of the immune system—the innate immune system including granulocytes, monocytes, macrophages, NK cells, and other cells of the innate immune system (including various dendritic cells, innate lymphoid cells, or mast cells) and the acquired (adaptive) immune system with T and B lymphocytes [77]. During SARS-CoV-2 infection, an inappropriately activated and dysregulated host immune response may cause immunopathology [78,79]. The course of infection may also be influenced by the external environment and the virus itself due to its genetic diversity, genetic evolution, and variable infectivity [77].

In the vast majority of patients, COVID-19 is asymptomatic or mild (80% of cases), but in some cases, there is a rapid and unpredictable course of the disease [77]. In about 5% of infected patients, the immune response runs uncontrollably, leading to inflammatory cascades, cytokine storms, and activation of coagulation cascades that further cause damage to other organs [60,80] and, in extreme cases, sepsis and death [81]. However, the detailed mechanisms of pathogenesis are not yet understood. A schematic of the COVID-19 run is shown in Figure 2.

## 3. Characteristics and Performance of KD

The ketogenic diet is a dietary regimen designed to increase the synthesis and utilization of ketone bodies [82,83]. Its characteristics are high fat and low carbohydrate. This macronutrient ratio allows for the production of nutritional ketosis—glucose-dependent tissues (heart and skeletal muscle) can partially adapt to fat metabolism and utilize ketone bodies such as acetone, BHB, or AcAc [84,85] as the leading energy substrate [84,86,87]. Ketosis is usually achieved by restricting carbohydrate intake below 50 g/day (5–10%) [82,83], providing 1–1.5 g/kg/day of protein, and 60 to 90% of energy coming from fat [23,88,89,90,91,92,93] (Table 1).

The concentration of ketone bodies in the circulation under normal physiological conditions usually fluctuates around 100 µM–1 mM [94] but can rise to 5–8 mM after a low-carbohydrate diet [86,95,96] and >20 mM in pathological conditions such as diabetic or alcoholic ketoacidosis [97].

**Table 1 viruses-15-01262-t001:** Composition of selected ketogenic diets to achieve nutritional ketosis.

Number	Type of KD	Carbohydrates /Day	Fat/Day	Protein/Day	Ketosis
1.	Classical Ketogenic Diet (CKD)—children [88,89,90]	10–15 g/day	90% EV	1 g/kg bw/day	ν
2.	Medium-Chain Trigliceryde (MCT) [91]	50 g/day 20% EV	78 g/day 70% EV	25 g/day 10% EV	ν
3.	Low-glycemic index treatment (LGIT) [98]	40 g/day 27% EV	60 g/day 45% EV	40 g/day 28% EV	ν
4.	Medium-Chain Triglyceride-Based Diet [92]				ν
5.	Modified Atkins Diet (MAD) [93]	10 g/day 5% EV	70 g/day 70% EV	60 g/day 25% EV	ν
6.	Mediterranean-ketogenic diet (MMKD) [23]	<20 g/day <10% EV	60–65% EV	30–35% EV	ν
7.	Eko Atkins diet [99]	26% EV	43% EV	31% EV	ν
8.	Ketonformula^®^, 50 g + usual diet [100]				ν

% EV—% of energy; bw—body weight; ν—yes.

Reduced by KD use, insulin levels together with high adrenaline levels lead to the release of free fatty acids and glycerol from adipocytes under the action of lipase. Free fatty acids undergo β-oxidation in the liver mitochondria with the production of acetyl-CoA, which under standard conditions is incorporated into the Krebs cycle by condensation of acetyl-CoA with oxaloacetate [94,97]. Low dietary carbohydrate supply increases the rate of fatty acid oxidation in the liver and consumes the hepatic pool of oxaloacetate for gluconeogenesis. Accumulated acetyl-CoA exceeds the ability to synthesize citrate thus becoming a precursor of ketone bodies [82,101,102].

The breakdown of fatty acids in the cytosol of prokaryotic cells and in the mitochondrial matrix of eukaryotic cells is preceded by the conversion of fatty acids in an ATP-mediated reaction catalyzed by acyl-CoA synthetase (so-called fatty acid thiokinase) into an active metabolite [103]. Once the acyl chains formed from fatty acids have passed through the mitochondrial membranes, they are introduced into the mitochondria via carnitine palmitoyltransferase (CPT-1) [104] and then broken down into acetyl-CoA via β-oxidation. Two acetyl-CoA molecules combine to form acetoacetyl-CoA (AcAc-CoA) in a reaction catalyzed by the enzyme thiolase, and then acetoacetyl-CoA reacts with the next acetyl-CoA molecule and with water to form 3-hydroxy-3-methylglutaryl-CoA (HMG-CoA) via the enzyme HMG-CoA synthase. HMG-CoA in the presence of a lyase is cleaved into acetyl-CoA and acetoacetate. In the mitochondrial matrix, acetoacetate is converted to acetone by non-enzymatic decarboxylation, or is reduced to β-hydroxybutyrate (β-OHB) via 3-hydroxybutyrate dehydrogenase [29,85,102].

The ketone bodies produced are released from the liver into the circulatory system. Acetone, as a volatile substance, does not convert back to acetyl-CoA and is removed from the body via the lungs and kidneys. After reaching extrahepatic tissues, β-hydroxybutyrate (BDH1) is converted to acetoacetate via the enzyme beta-hydroxybutyrate dehydrogenase, and acetoacetate is converted back to acetyl-CoA via the enzyme beta-ketoacyl-CoA transferase, goes through the citric acid cycle, and produces 22 ATP per molecule after oxidative phosphorylation (Figure 3) [85,102].

KD has been used to treat drug-resistant epilepsy for more than a century [17,18,19]. There is also a growing body of research supporting the efficacy of KD in neurodegenerative diseases [20,21,22], including the prevention of Parkinson’s and Alzheimer’s disease [23,24]. A body of evidence supports the beneficial effects of KD in the treatment of obesity [25,26,27,28], type 2 diabetes [26,28], cardiovascular disease [29,30,31], and cancer [28,32,33]. Several studies have also confirmed the immunostimulatory and anti-inflammatory effects of KD and ketone bodies [14,34,35,36,37,38,39] and their effects on the microbiome [105,106,107]. The experimental use of KD in mechanically ventilated patients in intensive care units has also yielded good results [108,109,110].

Recently, there has been a particular increase in research on KD related to the suggested role of KD and ketone bodies in preventing SARS-CoV-2 infection at both the viral entry and dissemination stages in the body. Studies in animals, human biological material, and humans have confirmed these suggestions on several key points. Thus, it appears that the use of KD and ketone bodies may be considered as an adjunctive intervention for the treatment of COVID-19, despite the fact that numerous studies indicate that COVID-19 infection alone can induce ketoacidosis, particularly in patients with diabetes [111,112,113,114], but also can cause a high increase in ketone bodies in patients without diabetes [111,115].

On the other hand, Karagannis et al. [116], in a study conducted in the peripheral blood of patients with moderate to severe COVID-19, found low BHB concentrations irrespective of calorie intake and blood glucose levels and thus concluded that ketogenesis is impaired in COVID-19 and may be mediated by a reduction in beta-hydroxybutyrate. However, in the light of the previously described reports, these conclusions need to be confirmed.

## 4. Efficacy of KD in COVID-19

### 4.1. Efficacy of KD/Ketone Bodies at the Stage of Virus Entry into the Cell

KD reduces ACE2 (angiotensin-converting enzyme 2) methylation and increases ACE2 mRNA expression in visceral adipose tissue and leukocytes, and decreases ACE2 and TMPRSS2 (transmembrane serine protease 2) levels in the lung.

The ACE2 used by SARS-CoV-2 to enter the cell is mainly found in the lung [117] but also in other locations including the small intestine, oesophagus, liver, colon [118], and adipose tissue [119,120].

The clinical course of infection depends largely on which host cells are infected first. Higher expression of the ACE2 gene in visceral and subcutaneous adipose tissue than in lung tissue may exacerbate infection, prolong hospitalization time, and increase the risk of death in obese patients with COVID-19 [121,122,123,124]. In a study by Izquierdo et al. [125], visceral adipose tissue and leucocyte samples from obese patients showed higher levels of ACE2 methylation and lower ACE2 gene expression than in normal-weight patients. The use of low-calorie KD reversed this unfavorable obesity-associated ACE2 methylation which resulted in an increase in ACE2 mRNA expression and reduced dangerous disease progression (Table 2).

Additionally, in a study in COVID-19-infected rats, the application of KD reduced the levels of ACE2 and TMPRSS2 (required to stimulate the S protein of the virus to allow it to enter the cell) in the lungs resulting in reduced susceptibility to COVID-19 infection [126] (Table 3). Thus, KD may act similarly to the cellular serine protease inhibitor TMPRSS2 approved for clinical use [127] which blocks SARS-CoV-2 entry and may be one option for treating the disease.

KD causes a shift of the renin–angiotensin–aldosterone system (RAS) from the AngI/ACE1/AngII/AT1 arm towards the counter-regulatory, anti-inflammatory arm of the RAS.

The renin–angiotensin–aldosterone system (RAS) plays a significant role in the regulation of the body’s water and electrolyte balance, as well as renal, neuronal, and endocrine functions related to cardiovascular control [128]. Due to the extensive and multi-organ action of the RAS, its overactivation contributes to the development of many diseases. The RAS can proceed through two pathways [129].

The first (classical) RAS axis includes the angiotensin-converting enzyme (ACE1), which converts angiotensin I (AngI) to AngII and, after binding to the type 1 receptor (AT1), can then induce vasoconstriction, water retention, and activation of the sympathetic nervous system [130]. The second axis involves angiotensin-converting enzyme 2 (ACE2), homologous to ACE1, converting AngI to angiotensin (1–9) and AngII to angiotensin (1–7) [131,132], which, via the G-protein-coupled Mas-R receptor [133,134], counteracts the effects of the classical RAS axis (Figure 4).

In the physiological state, both these axes remain in equilibrium. The SARS-CoV-2 virus infection induces internalization and degradation of ACE2 resulting in loss of Ang (1–7) and inhibition of its protective pathways [135,136], an imbalance of the systemic and local RAS, and its shift towards the unfavorable AngI/ACE1/AngII/AT1 axis [129,137,138]. Downregulation of ACE2 also induces a sustained elevation of Ang II via a local interaction with the AT1 receptor, triggering a vicious cycle in which Ang II decreases ACE2 levels, leading to further increases in local tissue Ang II levels [139]. Hypertension, diabetes, and coronary artery disease coexisting with SARS-CoV-2 infection trigger AngI/ACE1/AngII/AT1 overactivation or ACE2 deficiency [138]. In obesity, adipose tissue additionally secretes angiotensinogen, which can be converted to Ang II, leading to hyperactivation of the AngI/ACE1/AngII/AT1 arm and subsequent tissue damage [140].

Eira et al. [126] showed that the application of KD in obese rats resulted in a reduction in lung ACE2 levels greater than the application of a standard diet and a sucrose-enriched diet, which may be a counter-regulatory mechanism to mitigate the deleterious effects caused by RAS hyperactivation mediated by obesity. More so, the use of KD significantly reduced ACE1 protein content in lung tissue relative to rats fed a standard diet and reduced, although statistically non-significantly, AT1 protein content in this tissue (Table 2).

The use of KD via blockade of the classical RAS pathway mimicked the effects of ACE inhibitors and angiotensin II type 1 receptor blockers (ARBs) in patients with severe COVID-19, including a significant reduction in viral titers [141,142].

**Table 2 viruses-15-01262-t002:** Studies on the efficacy of KD/ketone bodies in SARS-CoV-2 on biological material from patients with COVID-19.

	Study Material/Quantity of Samples	Evaluation	Intervention/ Measurement	Results	Bibliography
1	Serum from patients with moderate COVID-19 or ARDS due to COVID-19/n = 155	Evaluation of the relationship between infection-induced metabolic changes and host immune responses in severe lung infections	Cultured human TCD4+/ E ketone ester BHB (d -β-hydroxybutyrate-(R)-1,3-butanediol monoester) added at days 0, 1, and 2 in an amount of 5 mM BHB (6 days of culture)	- in the peripheral blood of patients with moderate to severe COVID-19, low concentrations of BHB irrespective of calorie intake and blood glucose levels - BHB resulted in improved mitochondrial function, and fatty acid and amino acid oxidation; reduced glycolysis in CD4 + T lymphocytes - cultures of human TH1 CD4 + lymphocytes with BHB showed an increase in the number of CD4+ cells and IFNy production, and a decrease in PD-1 (programmed death receptor 1) expression	Karagiannis et al. [116]
2	Immune cells from patients with COVID-19; disease severity >IV^0^ and respiratory failure/n = 9/20	Evaluation of the efficacy of metabolic reprogramming of CD8 + T cells by ketone bodies in overcoming immune paralysis in patients with COVID-19	Peripheral blood mononuclear cells (PBMCs) from COVID-19 patients. Incubation with BHB, D/L-beta-hydroxybutyrate with a final concentration of 10 mM	- significant increase in granzyme B-expressing CD8 + lymphocytes and increased granzyme B expression per cell, - profound enhancement of CD8 + immune capacity - significantly increased secretion of the T lymphocyte cytokines CD8 + IFNγ, TNFα, perforin, and granzyme B - markedly increased cell lysis capacity of CD8 + T lymphocytes - CD8 + T lymphocytes showed significantly higher basal and maximal respiratory chain activity and better spare respiratory capacity - a tendency towards increased mitochondrial mass in CD8 + T cells - significantly increased mitochondrial ROS but cellular ROS levels remained unchanged	Hirschberger et al. [143]
3	DNA methylation levels of ACE2 from data sets generated by hybridization of subcutaneous (n = 32), visceral (n = 32) adipose tissue, and leukocyte samples (n = 34).	Evaluation of ACE2 methylation levels in different depots of adipose tissue and leukocytes in obesity	DNA methylation levels of ACE2. Data were compared based on degree of obesity and after 4–6 months of weight loss following weight loss therapy based on VLCKD vs. HCD vs. bariatric surgery	- decreased levels of ACE2 methylation and increased ACE2 mRNA expression in visceral adipose tissue and leukocytes in obese patients after VLCKD	Izquierdo et al. [125]
4	Simulated Kbhb antibody produced using in vitro chemical modification from a commercially obtained human serum antibody	Investigating the immunomodulatory mechanisms of β-hydroxybutyrate	Antibody solution with addition of free β-hydroxybutyrate	Kbhb did not affect the binding capacity of B38 to the S-protein of COVID-19 virus, indicating that the chemically modified antibody retained its original antigen-binding activity	Li et al. [144]

VLCKD—very low-calorie ketogenic diet, HCD—balanced hypocaloric diet, BHB—beta-hydroxybutyrate, ARDS—acute respiratory distress syndrome, ACE2—angiotensin-converting enzyme type 2, IFNy—interferon gamma, TNFα—tumor necrosis factor α, ROS—reactive oxygen species, Kbhb -.

**Table 3 viruses-15-01262-t003:** Studies on the efficacy of KD in SARS-CoV-2 in animal models.

	Study Design	Participants (n)/Age (year)	Evaluation	Intervention/ Measurement	Duration	Results	Bibliography
1	Random- ized clinical trial	Male C57BL mice infected with natural mouse beta coronavirus (mCoV) 6—mouse hepatitis virus strain A59 (MHV-A59) (reproducing the clinical features of COVID-19). Old mice (20–24 months) and young mice (2–6 months)	Effect of KD on the mouse defense response against MHV-A59 and identification of the underlying mechanism	Intranasal inoculation of mCoV-A59 into adult and old male mice. Feeding a standard vivarium feed (Harlan 2018s) or a ketogenic diet (Envigo, TD.190049)	5 days before infection	- protection against weight loss and hypoxemia caused by infection - improved survival of young but not old mice - significantly reduced mRNA expression of the Pro-inflammatory cytokines IL-1β, TNFα, and IL-6 in the lung, VAT, and hypothalamus - in old mice, a significant increase in cup cells, expansion of T ϒδ, and a significant decrease in subsets of proliferative cells - significant decrease in monocyte population subsets, change in monocyte compartiment, and loss of cluster with high levels of Chil3, Lmna, Il1r2, Lcn2, Cd33, and Cd24a; loss of monocyte subpopulation in cells with low interferon response - increase in T ϒδ cells, downregulation of TLR, Plk1, and aurora B signaling pathways in T ϒδ cells - in old mice, increased genes associated with reduced inflammation, increased lipoprotein remodeling - increased respiratory electron transport and complex I biogenesis in lung T ϒδ cells of old mice - reduced activation status of T ϒδ lymphocytes - decreased NLRP3 and caspase-1 mRNA in lung, visceral adipose tissue (VAT), and hypothalamus, and inflamasome activation in VAT - significantly reduced myeloid cell infiltration in the heart - BHB reduced oligomerization of ASC, which is an adaptor protein required for assembly of the inflamasome complex	Ryu et al. [42]
2	Random-ized clinical trial	Male albino rats of the Sprague strain/body weight 200–250 g	Measurements of ACE2, TMPRSS2, and RAS components and inflammatory genes in animal lungs and hearts	Chow diet (P: 27.0%, F: 13.0%, and C: 60. 0%), high-fat sucrose-enriched diet (P: 20.0%, F: 60.0%, and C:20.0%) or KD (P:20%, F: 80%, and C: 0%).	16 weeks	- reduced lung ACE2 and TMPRSS2 levels compared to a sucrose-enriched diet - KD in the lung resulted in decreased ACE1 and AT1 protein content, and a shift of the (RAS) system from the AngI/ACE1/AngII/AT1 arm towards the counter-regulatory, anti-inflammatory arm of the RAS KD induced anti-inflammatory effects—reduction of tlr4 and il6r gene expression, and a tendency to reduce tnfr1 gene expression	Eira et al. [126]
3	Random-ized clinical trial	K18-hACE2 mice aged 6–20 weeks infected intranasally with 60 PFU of SARS-CoV-2 (preclinical model of SARS-CoV-2 infection)	Assessment of T-cell metabolism and function	Supplementation with ketone ester 20 mg ml-1 (D- -hydroxybutyrate-(R)-1,3 butanediol monoester) in drinking water	8 days after infection	- decreased glucose dependence of CD4+ lymphocytes and increased the potential to oxidize amino acids and fatty acids thereby increasing their ability to produce IFNy and antiviral protection - ketone ester promoted faster recovery from weight loss and reduced lung injury, resulting in improved overall survival	Karagiannis et al. [116]

KD—ketogenic diet, P—protein, F—fat, C—carbohydrates, VAT—visceral adipose tissue, ER—endoplasmic reticulum, NLRP3—NLRP3 inflammasome, BHB—beta-hydroxybutyrate, TMPRSS2—transmembrane serine protease 2, RAS—renin-angiotensin-aldosterone system, ACE1—angiotensin-converting enzyme I protein, AT1—angiotensin receptor type 1 protein.

### 4.2. Effectiveness of KD/Ketone Bodies in Regulating Innate and Acquired Immunity

The atypical course of COVID-19 in about 20% of patients may be due to a unique dysregulation of the immune response [79]. In contrast, ketone bodies may regulate immune cells, although their effect on the immune response is not fully understood [34,36].

Previous studies suggest that BHBs, by preventing metabolic reprogramming associated with COVID-19 infection and improving mitochondrial function, resulted in reduced glycolysis in CD4+ lymphocytes, improved respiratory chain function, and could provide an alternative carbon source to OXPHOS. In studies with animal models and human biological material, it was found that the use of KD/ketone bodies could support the host immune response as shown in Figure 5.

#### 4.2.1. BHB Reduces Cellular Dependence on Glucose

In healthy organisms, when immune cell activation is not required, glucose is metabolized by the glycolytic pathway of primary importance in sustaining life in almost all organisms, yielding pyruvate and NADH (nicotinamide adenine dinucleotide) as products [145]. These intermediates are then utilized in the mitochondria, in the tricarboxylic acid (TCA) cycle of cellular respiration, and oxidative phosphorylation (OXPHOS), forming intermediate compounds that are used in the electron transport chain to produce 32 molecules of adenosine triphosphate (ATP) for every molecule of glucose [146,147]. During the immune response when immune cells are activated, for example, in COVID-19 infection, effector T cells to meet the increased bioenergetic demands undergo metabolic reprogramming involving a switch from oxidative phosphorylation to glycolysis [148]. Such reprogramming results in compromised ATP production by OXPHOS [145].

A study by Karagannis et al. [116] found that T lymphocytes in bronchoalveolar lavage fluid (BALF) and blood from subjects with SARS-CoV-2-induced acute respiratory distress syndrome (ARDS) showed a significantly altered metabolic profile towards glycolysis, but still exhibited a residual capacity to oxidize fatty acids and amino acids [116]. However, in COVID-19, the available amount of amino acids and BHB is reduced [116,149], and marked changes in lipid and amino acid metabolism are also observed [150]. Meanwhile, BHB treatment resulted in improved mitochondrial functionality for fatty acid and amino acid oxidation while leading to reduced glycolysis in CD4+ lymphocytes in patients with severe COVID-19 [116]. Similarly, the use of ketone ester in a preclinical mouse model of SARS-CoV-2 infection reduced the dependence of CD4+ lymphocytes on glucose and increased the potential to oxidize amino acids and fatty acids thereby increasing their ability to produce interferon-γ (IFNy), which plays an important role in the body’s immune response during infection control [151] and antiviral protection [116].

#### 4.2.2. BHB Promotes Mitochondrial Function, Improves Respiratory Chain Function, and May Provide an Alternative Carbon Source to OXPHOS

SARS-CoV-2 virus RNA localizes to mitochondria and then manipulates mitochondrial function causing mitochondrial dysfunction [152], disrupting platelet mitochondrial respiratory chain function and mitochondrial oxidative phosphorylation (OXPHOS) [153,154], which in turn will disrupt T-cell function [148,155] and may be the cause of defective adaptive immune responses in COVID-19.

Ketone bodies through immunometabolic reprogramming in T cells can be used for energy production via mitochondrial oxidative phosphorylation (OXPHOS) [97,156].

Hirschberger et al. [143] demonstrated the efficacy of using BHB to increase the energy capacity of CD8 + cells during COVID-19. CD8 + lymphocytes under the influence of BHB showed significantly higher basal and maximal respiratory chain activity and better respiratory reserve capacity, indicating enhanced energy production in the mitochondria. BHB caused an increase in mitochondrial mass in CD8 + lymphocytes. Thus, they demonstrated that T lymphocytes were able to utilize BHB via oxidation in the Krebs cycle, which drives OXPHOS with excellent efficiency [157,158] and showed that ketone bodies direct human CD8 + T lymphocytes towards aerobic mitochondrial metabolism during COVID-19, thus enabling improved energy supply [143].

These observations corroborate an earlier study by Karagannis et al. [116], who demonstrated on cultured mouse and human TH1 CD4+ lymphocytes the ability of BHB to enhance CD4+ mitochondrial functionality and the role of BHB in supporting OXPHOS in mouse CD4+ lymphocytes. In the same study, Kragannis et al. [116] showed that providing mice with a ketogenic diet resulted in metabolic reprogramming of pulmonary CD4+ cells towards OXPHOS and a reduction in their glycolytic capacity, and that BHB provides an alternative carbon source to drive OXPHOS and the production of bioenergetic amino acids (glutamate and aspartate) and glutathione in CD4+ T cells, which is important for maintaining redox balance.

#### 4.2.3. BHB Prevents CD4+ T-Cell Dysfunction

Adequate activation for proliferation, clonal expansion, and effector function of cytotoxic CD4+ type 1 helper T cells (H1 T cells) is crucial in clearing viral infections due to their ability to produce cytokines including interferon-γ (IFNγ) [159,160] and their modulation of the activity of other antibodies [116,161].

Severe COVID-19 is associated with significantly reduced CD4+ lymphocyte frequency and dysfunction [162,163,164,165]. This is caused by direct binding of the SARS-CoV-2 peak glycoprotein to the CD4+ molecule, which facilitates the entry of SARS-CoV-2 into helper T cells, and further causes cellular impairment or death [166].

In a study by Karagiannis et al. [116], when human TH1 CD4 + lymphocytes obtained from patients with severe COVID-19 with BHB were cultured, an increase in CD4+ cells and IFNy production, and a decrease in PD-1 (programmed death receptor 1) expression were observed. Additionally, previous studies conducted on human immune cells indicated that BHB led to a significant transcriptional upregulation of CD4+ cell cytokines, interleukin (IL)2, IL4, IL8, and IL22 which improved the immune capacity of human T cells [156]. However, according to a study by Ryu et al. [42] in an animal model, ketogenesis in infected old mice did not affect the frequency of CD4+ and effector memory CD8+.

#### 4.2.4. KD and Activation of Ketogenesis Reduced the Number of Pathogenic Monocytes in the Lung, Blocked Infiltration of a Pathogenic Subset of Monocytes in the Lung, and Resulted in the Loss of a Subpopulation of Monocytes with Low Interferon Expression

Monocytes, representing 15–30% of circulating lymphocytes [167], play an important role in both innate and adaptive immunity, inflammation, and tissue remodeling [168]. During infection, they escape from the blood vessels into the surrounding tissues and, after being transformed into macrophages along with them, constitute sentinel cells, engulf viruses, and coordinate the immune system [169]. However, an uncontrolled increase in monocyte levels leading to inflammation has been observed in patients with severe COVID-19 [170].

Ryu et al. [42], in a study with mice (Table 2), observed a significant decrease in subsets of the monocyte population in old, infected mice after KD injection. Induction of ketogenesis in mice by KD application also blocked infiltration of a pathogenic subset of monocytes in the lungs, which show high expression of S100A8/9, an immune-activating alarmin [171,172,173]. Uncontrolled production or sustained expression of alarmins can be dangerous to the host by inducing a cytokine storm or excessive inflammation [174,175], as observed in severe cases of COVID-19 [78,176,177]. In addition, they noted a loss of a subpopulation of monocytes with low interferon expression after KD treatment [42]. This suggests the induction of an immune response following ketogenesis in infected mice, quenching the overexpression of alarmins and thereby preventing the formation of excessive inflammation and/or cytokine storm in COVID-19.

#### 4.2.5. KD Caused a Slight Decrease in Interleukin-6 (IL-6) Levels in Humans and Reduced mRNA Expression of the pro-Inflammatory Cytokines IL-1β, TNFα, and IL-6, the Inflammatory Genes TLR4 and ILR6, and Suppressed TNFR1 Gene Expression in Mice

Numerous studies have shown that severe and critically ill COVID-19 patients experience a ‘cytokine storm’ associated not only with excessive cytokine production, but with dysregulation of immune cell function and systemic inflammation [178]. In such patients, a slight decrease in interleukin-6 (IL-6) levels was observed following a eucaloric ketogenic diet (EKD) [14] (Table 4).

In a study by Ryu et al. [42], older mice fed a KD diet after subtotal infection with mCoV-A59 showed significantly reduced mRNA expression of the pro-inflammatory cytokines IL-1β, TNFα, and IL-6 in the lung, visceral adipose tissue (VAT), and hypothalamus. The use of KD also resulted in reduced expression of the inflammatory genes TLR4 and ILR6, and suppressed expression of the TNFR1 gene in the lungs of mice compared to a sucrose-enriched diet [126] (Table 2). Thus, KD may, by affecting signaling pathways, improve outcomes in COVID-19, including by blocking the expression of genes encoding pro-inflammatory cytokines.

Interestingly, some patients with COVID-19 in severe condition expressed low levels of cytokines in the blood resulting from immune cell depletion [79,181]. In functional studies by Hirschberg et al. [143] on cells obtained from COVID-19 patients (Table 3), incubating them with BHB resulted in increased TNF-α expression. This improved the CD8 + immune capacity, which may consequently counteract the severe complications of COVID-19. However, given the results of studies demonstrating the adverse effects of excess TNF-α in COVID-19, these reports should be approached with caution. While this effect of BHB would be beneficial in cases of immune cell depletion, it could already have adverse consequences in other patients.

#### 4.2.6. KD and Activation of Ketogenesis Resulted in Increased Homeostaticity of T γδ Lymphocytes

During viral infection, Tγδ lymphocytes are also activated, which, although a small proportion of the total peripheral blood T cells, have the ability to respond rapidly to stimuli, making them one of the most important innate and acquired lines of defense against viral infections [182,183]. Increased activation of Tγδ lymphocytes was observed in patients with COVID-19, they were also more differentiated than control samples [184] and quantitatively reduced levels were observed compared to healthy patients [185,186]. Such an abnormal Tγδ profile in COVID-19 favoured a worse disease scenario [182].

Meanwhile, Ryu et al. [42] showed an increase in Tγδ cells and an increase in T ϒδ lymphocyte expansion in the lungs of aged mice as well as a reduced Tγδ lymphocyte activation state after application of KD to mCoV-A59-infected male mice reproducing the clinical features of COVID-19 for 5 days prior to infection. Application of KD resulted in a significant increase in genes associated with reduced inflammation, increased lipoprotein remodeling and downregulation of TLR, Plk1, and aurora B signaling pathways in Tγδ lymphocytes. Lung Tγδ lymphocytes from mCoV-A59-infected mice treated with KD showed increased respiratory electron transport and complex I biogenesis. In addition, Golgi retrograde transport to the ER and cell cycle were downregulated, suggesting a reduced activation status of ϒδ T cells. The results suggest that Tγδ T cells proliferated with KD are functionally more homeostatic and immunologically protective against mCoV-A59 infection mimicking SARS-CoV-2. This confirmed previous observations associated with the use of KD in influenza A virus infection, resulting in the expansion of Tγδ cells in the lungs of mice, improved immune barrier function, and lung barrier integrity [187,188,189], and thereby increased antiviral resistance. Importantly, Tγδ cell expansion required metabolic adaptation to KD, whereas feeding mice a high-fat, high-carbohydrate diet or providing ketone bodies did not protect against infection [34].

#### 4.2.7. KD and Ketogenesis Activation Inactivates the NLRP3 Inflammasome

The NLRP3 inflammasome [190,191], an essential component of the innate immune system, is hyperactivated in response to SARS-CoV-2 infection [40,192]. Hyperactivation of the NLRP3 inflammasome results in increased levels of caspase-1 responsible for inducing the secretion of pro-inflammatory cytokines (IL-1β, IL-18, IL-6, and TNF-α) [191,193,194]. Abnormal activation of the NLRP3 inflammasome in COVID-19 is involved in the pathogenesis of cytokine storm [190,195], acute respiratory distress syndrome (ARDS) [193,196], and acute lung inflammation (ALI) [197], as well as pathological multi-organ damage [194,198].

Ryu et al. [199], in a study with mice and mouse macrophages after mCoV-A59 infection replicating the clinical features of COVID-19, observed that administration of a ketogenic diet for 5 days in an aging mouse model of mCoV-A59 infection, significantly decreased NLRP3 mRNA and caspase-1 in the lung, visceral adipose tissue (VAT), and hypothalamus, and inflammasome activation in the VAT. BHB also reduced oligomerization of ASC, an adaptor protein required for the assembly of the inflammasome complex. This confirms previous observations by Youm et al. [36], who showed in a mouse model of inflammatory disease that BHB caused mechanistic inhibition of NLRP3 inflammasome activation and also reduced interleukin (IL)-1β and IL-18 production in human monocytes. In vivo, BHB or a ketogenic diet attenuated caspase-1 activation and IL-1β secretion in mouse models of diseases mediated by NLRP3.

Thus, the NLRP3 inflammasome may be a potential target for the treatment of COVID-19 [191,194], which is strongly supported by studies showing improved saturation, reduced hospitalization time and reduced mortality in patients with COVID-19 after treatment with NLRP3 inflammasome inhibitors such as Tranilast [200] or colchicine [198]. The anti-inflammatory effect of metformin in COVID-19 was also associated with inhibition of NLRP3 inflammasome activation [196].

#### 4.2.8. BHB Prevents CD8+ T-Lymphocyte Dysfunction

CD8+ T lymphocytes contribute to protective immune responses against SARS-CoV-2 [201]. They play an essential role in controlling viral infection by lysing virus-infected cells and producing effector cytokines [202]. However, in severe COVID-19, profound dysfunction of CD8+ lymphocytes has been observed involving their over-activation, reduced absolute number, and functionality [201,203,204,205,206,207,208].

In a study by Hirschberger et al. [143] on immune cells from patients with severe COVID-19 infection, application of BHB resulted in an increase in CD8+ lymphocytes and a significant enhancement of CD8+ immune capacity, and increased secretion of CD8+ T-cell cytokines (IFNγ, TNFα, perforin, and granzyme B). They also showed that CD8+ lymphocytes had a significantly increased capacity for cell lysis after BHB treatment. Thus, administration of BHB significantly enhanced the CD8+ T-lymphocyte functions impaired by SARS-CoV-2 infection in severely ill patients. In addition, under the influence of BHB, there was a tendency to increase mitochondrial mass in CD8+ T cells and redirect CD8+ T cells towards aerobic mitochondrial metabolism allowing for better energy supply. This reprogramming of CD8+ cells by BHB may be a promising strategy to overcome immune paralysis in patients with COVID-19. These findings confirmed previous observations on primary human cells and from an interventional immunological study with healthy volunteers, where both ketone bodies and a ketogenic diet prevented CD8+ T-cell dysfunction through immunometabolic reprogramming [156].

#### 4.2.9. BHB and KD Prevent Cell Death via the Pseudo Receptor Pathway (by Perforin and Granzymes)

In the peripheral blood mononuclear cells of patients with severe COVID-19, in addition to CD8+ lymphocyte dysfunction, significantly higher granzyme B and perforin expression was observed in CD8+ lymphocytes compared with mild disease [162]. Incubation of peripheral blood mononuclear cells obtained from patients with high COVID-19 severity and respiratory failure with BHB resulted in a significant increase in granzyme B-expressing CD8+ lymphocytes, increased granzyme B expression per cell, significantly increased secretion of perforin and granzyme B, and high cell lysis capacity [143]. This demonstrates the enhanced ability of CD8+ lymphocytes after BHB treatment to eliminate the virus from the host [209,210,211,212].

Perforins polymerize to form a channel that allows the free flow of ions and small polypeptides into the cell. The result is cell instability, impaired osmoticity, loss of energy and destruction of genetic material and, ultimately, pseudo receptor-mediated cell destruction [213].

In the absence of perforin, high viral titers activate CD8+ lymphocytes, maintain high CD8 T-cell numbers for long periods of time, and produce high levels of IFN-y but are unable to clear the infection [214]. Cunningham et al. [212] suggested a link between perforin expression and COVID-19 resistance.

The role of granzymes, on the other hand, is to activate perforin polymerization, penetrating through perforin channels into the target cell, where they degrade integral structure proteins, cytosolic proteins, and cell organelles. They also bind to and degrade chromatin proteins. This leads to the exposure of DNA to endonucleases and its destruction [213].

#### 4.2.10. BHB as a Substrate for Histone Kbhb

Xie et al. [215] showed that β-Hydroxybutyrate acts as a substrate for histone Kbhb (a post-translational modification of the protein), which then activates gene expression, independent of acetylation. In an unreviewed study on simulated Kbhb antibodies produced using in vitro chemical modification from a commercially obtained human serum antibody treated with β-Hydroxybutyrate, it was shown that Kbhb did not affect the binding capacity of B38 to the S protein of COVID-19 virus, indicating that the chemically modified antibody retained its original antigen-binding activity [144].

### 4.3. BHB Increased Mitochondrial mROS, but without Uncontrolled ROS Expansion

In many pathological conditions, increased mitochondrial oxidative phosphorylation translates into an increase in mitochondrial reactive oxygen species (mROS) [216,217]. At the same time, mROS are very important for the homeostasis of the body [218] including an adequate T-cell immune response [219]. Nine times higher ROS production was observed in patients with COVID-19 compared to healthy subjects, and was particularly high in patients requiring mechanical ventilation [220].

Hirschberger et al. [143], in a study on cells obtained from COVID-19 patients incubated with BHB, found that mitochondrial ROS were significantly elevated after incubation with ketone bodies, indicating an increased immune capacity of CD8 + cells. At the same time, cellular ROS levels remained unchanged, indicating that there was no uncontrolled ROS expansion that could impair cell viability.

### 4.4. KD in Animal Models Resulted in Protection against Weight Loss and Hypoxemia, Faster Recovery, Reduced Lung Damage, and Resulted in Better Survival of Young Mice

In animal studies, Ryu et al. [42] also showed that KD-fed mice infected with mCoV-A59 were protected against weight loss and hypoxemia caused by the infection. Compared to the control group, young mice infected with a lethal dose of mCoV-A59 (PFU 1e6) and fed KD before infection showed better survival in contrast to old mice, which were not protected from death by KD. KD also significantly reduced myeloid cell infiltration in the heart.

Karagannis et al. [116], in a study with mice in a preclinical model of SARS-CoV-2 infection, found faster recovery from weight loss due to infection and also reduced lung damage, resulting in increased overall survival when mice were fed KD or ketone ester.

### 4.5. KD in Humans Increased Survival, Reduced the Need for Hospitalization for COVID-19, and Showed a Protective Role against Metabolic Abnormalities after COVID-19

Previous studies have shown that the use of KD resulted in a milder course and lower mortality in patients with COVID-19 and also reduced obesity.

Sukkar et al. [14] in a pilot study showed higher survival and significantly less need for ICU hospitalization in COVID-19 patients following a eucaloric ketogenic diet (EKD 1800–2100, carbohydrates <30 g, 5–6% of total energy) from admission compared with patients following a standard diet. The results of this study indicated for the first time the role of clinical KD-based nutritional therapy as a pathophysiological support for drug therapy in improving the prognosis of COVID-19. The EKD used proved to be safe and no adverse effects were observed.

In a retrospective study, Volk et al. [180] demonstrated a significant reduction in body weight in patients with type 2 diabetes and obesity who used KD (5–10% energy in the form of carbohydrate, 15–20% protein, and 70–75% fat) before COVID-19 diagnosis and during treatment. Both obesity and type 2 cu-diabetes predispose to a more severe course of COVID-19. Obesity through common pathophysiological features with COVID-19, such as chronic inflammation, immune dysregulation, oxidative stress, increased cytokine production, and endothelial dysfunction worsens the course of the disease. An additional increased number of adipocytes express ACE2 and increased adipose tissue constitutes a larger viral reservoir [44,221]. The introduction of KD through weight reduction resulted in reduced inflammation and thus a lighter course of COVID-19 and less need for hospitalization. KD, by limiting carbohydrate supply and reducing glucose and insulin levels, may also have reversed the dysfunction of the immune response that increases the risk of spreading invasive pathogens in patients with diabetes [222].

Reductions in body weight, body mass index (BMI), amount of body fat with preservation of lean mass, and improvements in muscle strength and physical performance were observed by Camajani et al. [179] in a woman with sarcopenic obesity (SO) and dyslipidemia, following hospitalization for bilateral interstitial pneumonia due to COVID-19, using VLCKD (800 kcal/day, carbohydrate 28 g (14.6%), fat 35 g (38.7%), and protein 85 g (46.7%)) combined with interval training for 6 weeks after treatment. Given that SO exacerbates the decline in skeletal muscle mass and strength, in turn, the reduced muscle mass is significantly associated with the occurrence of complications after COVID-19 [223], and the use of VLCKD reversed the expected adverse outcomes.

In the same patient with SO, a decrease in blood pressure, insulin, fasting glycemia, HOMA index, triglycerides, and LDL cholesterol was observed with a concomitant increase in HDL cholesterol [179]. These findings suggest a protective role of KD against metabolic abnormalities after COVID-19.

Previous studies have indicated beneficial metabolic effects of KD use, with COVID-19 diabetes (type 1, type 2) [224,225], hypertension, and cardiovascular disease [1,225] being associated with an increased risk of severe disease and higher mortality [224]. Thus, improving parameters related to the course of these diseases may reduce the severity of the course and mortality of COVID-19 [226,227,228].

### 4.6. Future Research Directions

Many years of research using ketogenic diets to treat epilepsy, obesity, and type 2 diabetes have demonstrated the safety of KD. However, they do have some limitations as clinical interventions, including the difficulty in implementing large-scale dietary changes and strict patient adherence to these recommendations. Despite the encouraging results, it is important to note the limitations of the studies analyzed such as the small sample size and lack of randomized controlled trials, which may affect the generalizability of the results. Furthermore, animal studies should be extrapolated to humans with great caution.

To confirm the efficacy of KD/ketone bodies in the prevention and treatment of COVID-19, randomized clinical trials involving COVID-19 patients are needed, with assessment of the duration and severity of the course of infection and patient mortality. These trials should analyze the outcomes of patients at different stages of COVID-19 disease progression following the introduction of KD or ketone bodies therapy, which would identify the time at which such an intervention would be most effective and eliminate groups of patients at risk of developing ketoacidosis.

Soliman et al. [43], to support the treatment of SARS-CoV-2, suggested altering host lipid metabolism through the use of KD combined with fasting consisting of a ketogenic breakfast combined with medium-chain fatty acid (MCT) supplementation from coconut oil, a lunch with MCT inclusion, an 8–12 h fast and a dinner rich in fruit and vegetables. Cooper et al. [229] in a study involving patients with COVID-19 and a history of hyperglycemia and/or hyperinsulinemia proposed an algorithm of restricted carbohydrate management, administered either enterally or parenterally, restricting carbohydrates and intravenous solutions containing dextrose in the diet to a minimum. Restriction of the introduced diet would be dependent on continuously monitored blood glucose, insulin, and ketone bodies. The algorithm should include supplementation with vitamin D, magnesium, and zinc.

Although the results obtained to date do not allow a clear recommendation for the use of KD and/or ketone bodies for the treatment of COVID-19, a model of nutritional management using KD can be proposed, which would involve a low-carbohydrate diet that increases blood ketones throughout the course of the disease, while monitoring patient outcomes and adjusting carbohydrate restriction according to the course of the disease. This would maintain normal blood glucose levels and stimulate immune cell function in the early stages, reduce glucose metabolism and associated pro-inflammatory signaling during symptom severity, and promote anti-inflammatory processes when respiratory difficulties arise. Regardless of the timing of the introduction of KD or ketone bodies therapy, it could have the effect of reducing the incidence of progression to ARDS, protecting organs from oxidative and inflammatory damage, and shortening the duration of the disease. The use of KD could also have a positive effect in patients who have survived severe ARDS causing psychiatric problems and chronic fatigue [3] due to the mitigation of cell death and tissue damage.

## 5. Conclusions

There is a wealth of scientific evidence on the effectiveness of nutritional interventions in disease prevention and treatment. It is also known that a good nutritional status of the body has a great impact on the course of COVID-19. However, to date, the best nutritional approach for use during the SARS-CoV-2 pandemic has not been established. This review examines the evidence on the areas of efficacy of the use of ketogenic diets or ketone bodies for prevention of or during COVID-19 infection in animal models, in humans, and in tissue models. The efficacy of KD or ketone bodies at the stage of SARS-CoV-2 virus entry into the cell has been shown to result from (a) KD decreasing ACE2 methylation and increasing ACE2 mRNA expression in visceral adipose tissue and leukocytes, downregulation of ACE2 and TMPRSS2 key elements used by SARS-CoV-2 to infect cells in the lung, and (b) a shift of the renin–angiotensin–aldosterone system (RAS) from the AngI/ACE1/AngII/AT1 arm towards the anti-inflammatory RAS arm. The efficacy of KD/ketone bodies in the regulation of innate and acquired immunity was also confirmed by (a) BHB reducing cellular dependence on glucose, (b) BHB promoting mitochondrial function and improving respiratory chain function, which could become an alternative carbon source to OXPHOS, (c) BHB preventing CD4+ and CD8+ T-lymphocyte dysfunction, (d) KD and activation of ketogenesis reducing the number of pathogenic monocytes in the lungs, (e) KD inducing a slight decrease in IL-6 levels in humans and decreased mRNA expression of the pro-inflammatory cytokines IL-1β, TNFα, and IL-6, the inflammatory genes TLR4 and ILR6, and inhibiting TNFR1 gene expression in mice, (f) increased homeostaticity of T γδ lymphocytes, (g) inactivation of the NLRP3 inflammasome, and (h) prevention of cell death via the pseudo receptor pathway (via perforin and granzymes). At the same time, BHB increased mitochondrial mROS, but without uncontrolled ROS expansion. In animal models, KD resulted in protection against weight loss and hypoxemia, faster recovery, reduced lung damage, and better survival of young mice. In humans, KD also increased survival, reduced the need for COVID-19 hospitalization, and showed a protective role against metabolic disorders after COVID-19.

In conclusion, it can be stated that the multidirectional beneficial effects of KD and ketone bodies pave the way for research into metabolic treatment in COVID-19 [4]; however, the selection of KD effective in the prevention and/or treatment of COVID-19 is fraught with problems and requires further studies that would not only confirm its efficacy but also identify the best time to apply it during infection, define the KD model, and identify contraindications for the use of such an intervention against COVID-19.

## Figures and Tables

**Figure 1 viruses-15-01262-f001:**
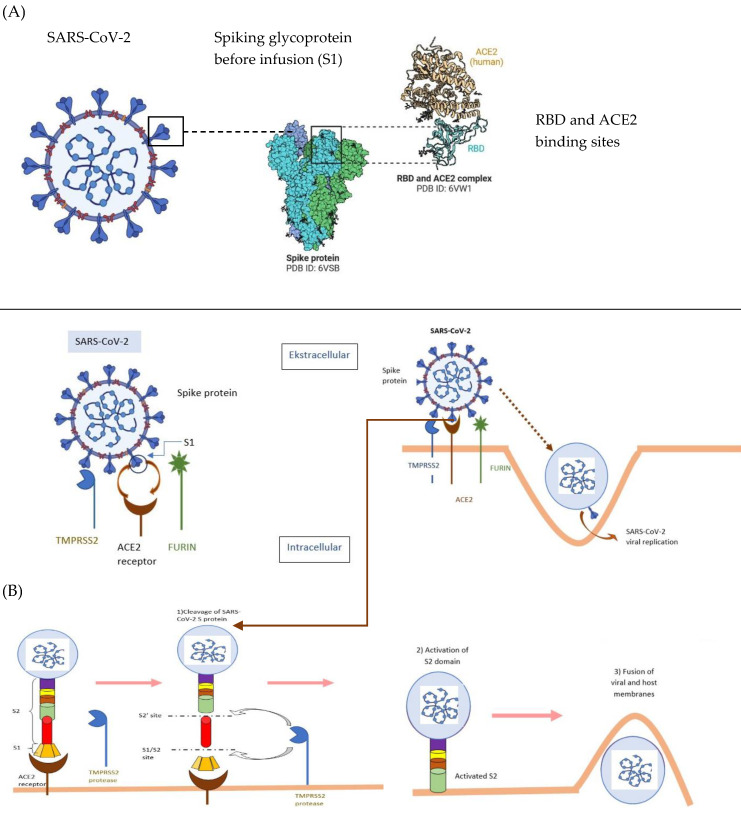
Interaction between the SARS-CoV-2 coronavirus spike protein and the host cell receptor ACE2 [72]. RBD—receptor; ACE2—angiotensin-converting enzyme 2; S1-, S2-, S2′- subunits of Spike (S) structural protein; TMPRSS2—Trans-membrane serine protease 2; Schematic of SARS-CoV-2 entry into the host cell. (**A**) presentation of protein S; (**B**) binding of protein S into S1 and S2 sites occurs. The activated S2 domain assists in the fusion of the SARS-CoV-2 molecule with the cell membrane. (Figure was created using templates available at BioRender.com, accessed 21 May 2023).

**Figure 2 viruses-15-01262-f002:**
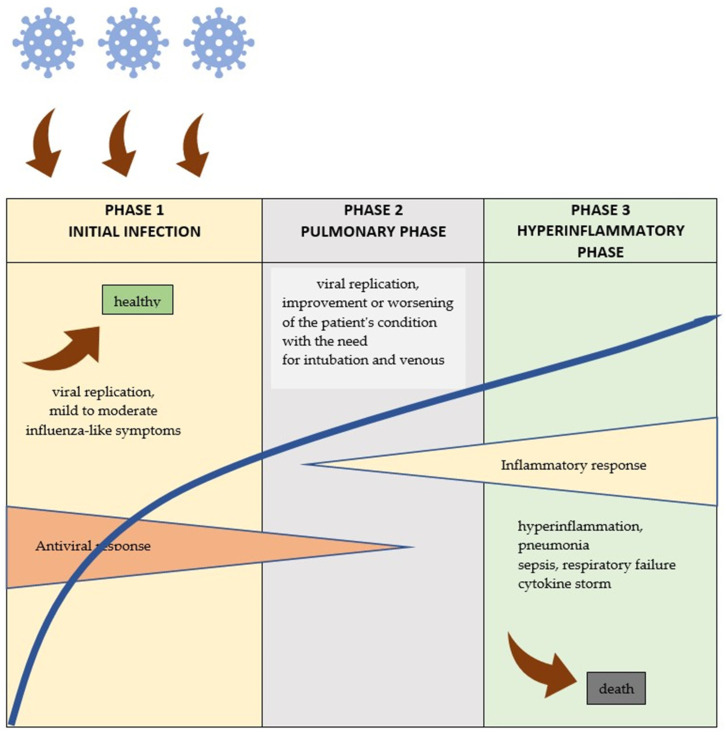
COVID-19 flow diagram [63].

**Figure 3 viruses-15-01262-f003:**
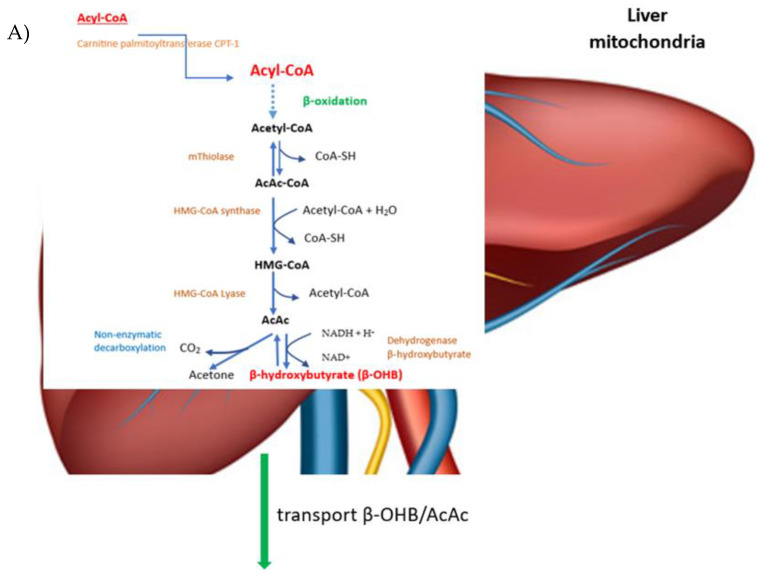
Scheme of metabolism of ketone bodies. BDH1—D-β-OHB dehydrogenase, SCOT—succinyl-CoA:3-oxoacid-CoA transferase, TCA—the tricarboxylic acid cycle. (**A**) The ketone bodies acetoacetate (AcAc), β-hydroxybutyrate ((β-OHB), and acetone are majorly produced in the mitochondrial matrix of liver cells during the oxidation of fatty acids. Ketone synthesis begins in the mitochondria of the liver after transporting the fatty acyl CoA molecule into the inner mitochondrial membrane by the carnitine shuttle. The transmembrane protein to move fatty acyl CoA molecules across the mitochondrial membrane is carnitine palmityl transferase I (CPT-1) The fatty acyl CoA molecules undergo beta-oxidation to become acetyl CoA molecules. Acetyl CoA molecules are either converted to malonyl CoA by acetyl CoA carboxylase or acetoacetyl CoA by 3-ketothiolase. Acetoacetyl CoA is further converted to 3-hydroxy-3-methylglutaryl CoA (HMG CoA) by HMG CoA synthase. HMG CoA synthase is the rate-limiting step for the synthesis of ketone bodies. HMG CoA is finally converted to acetoacetate by HMG CoA lyase. (**B**) Upon arrival to the mitochondria of distant organs, ketone bodies become utilized for energy. The first step involved is an enzyme that converts acetoacetate to acetoacetyl CoA. This enzyme is called succinyl CoA-oxoacid transferase (SCOT), and it is the rate-limiting step for the utilization of ketones for energy. High concentrations of acetoacetate feedback negatively on SCOT to decrease ketone conversion. Finally, acetoacetyl CoA is converted to acetyl CoA by methylacetoacetyl CoA thiolase. Acetyl CoA may be turned into citrate and churned through the citric acid cycle to produce FADH2 and NADH, or it can be converted to oxaloacetate and used in gluconeogenesis. Abbreviations: BDH1—D-β-OHB dehydrogenase, SCOT—succinyl-CoA:3-oxoacid-CoA transferase, TCA—the tricarboxylic acid cycle.

**Figure 4 viruses-15-01262-f004:**
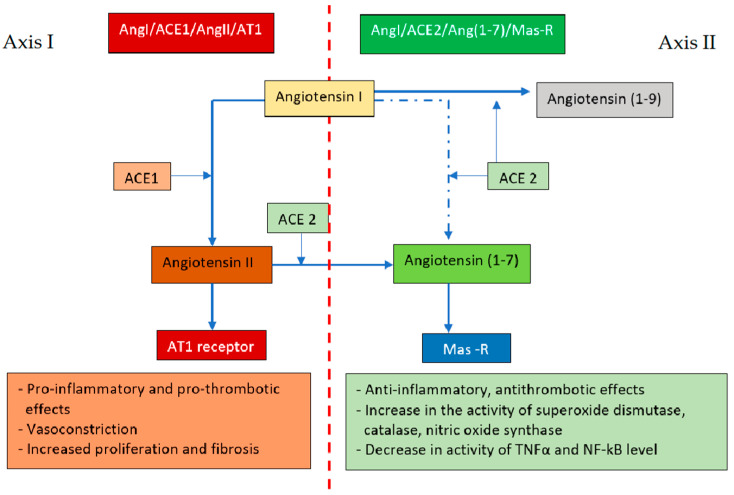
Opposing actions of angiotensin II and angiotensin (1–7). ACE1—angiotensin-converting enzyme, ACE2—angiotensin-converting enzyme 2, Mas-R—Mas-R receptor. Axis I: ACE1 converts AngI to AngII and, after binding to the AT1 receptor, can cause vasoconstriction, water retention, and sympathetic nervous system activation; Axis II: ACE 2 converts AngI to angiotensin (1–9) and AngII to angiotensin (1–7) which, via the G-protein-coupled Mas-R receptor, counteracts the effects of Axis I.

**Figure 5 viruses-15-01262-f005:**
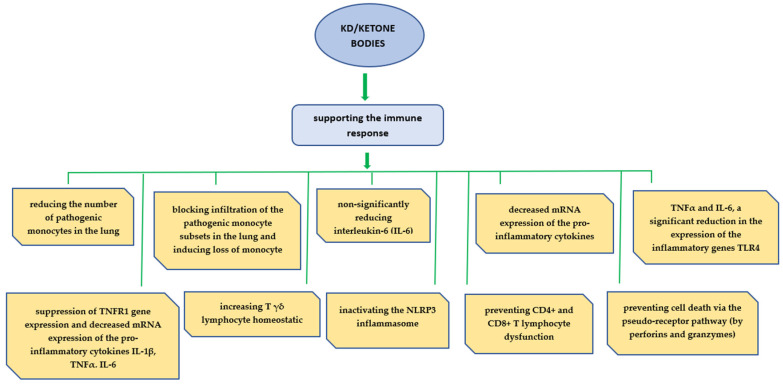
The role of the ketogenic diet and ketone bodies in immunity. The KD expands metabolically protective γδ T cells that restrain inflammation. β-hydroxybutyrate inhibits NLRP3 inflammasome activation. γδ T cells in visceral adipose tissue are activated by KD and might have important roles in adipose tissue remodeling and homeostasis. HIF-1α also induced expression of the proinflammatory cytokines TNF-α and IL-6 and decreased their mRNA expression. BHB improved respiratory chain function and prevented CD4+ and CD8+ T-lymphocyte dysfunction.

**Table 4 viruses-15-01262-t004:** Studies on the efficacy of KD in COVID-19 in humans.

	Study Design	Participants (n)/Age (year)	Evaluation	Intervention	Duration	Results	Bibliography
1	Retrospective study in hospitalized patients with COVID-19	EKD n = 34 vs. ESD n = 68	30-day mortality, admission to the intensive care unit; effect of ECD on biological and inflammatory parameters, especially IL-6	ESD 1800–2100 kcal, C: 42–50%, P: 16–20%, F: 26–30%; EKD 1800–2100 kcal, C <30 g, 5–6%, P: 17–18%, and polyunsaturated/unsaturated/saturated fat ratio 3:2:1	follow the diet for a mini-mum of 7 days	- statistically significantly higher survival rates - lower need for intensive care unit hospitalization - reduction in IL-6 between days 0 and 7	Succar et al. (2021) [14]
2	Case report—COVID-19- positive woman, hospitalized for bilateral pneumonia, with sarcopenic obesity, hypertension, hyperinsulinemia, hypercholesterolemia, and hypertriglyceridemia	n = 1, age 55	Anthropometric parameters and body composition, muscle mass and strength, time required to rise from a sitting position without the use of hands, blood pressure, insulin and fasting blood glucose levels, creatinine, triglycerides, LDL cholesterol, HDL cholesterol, vitamin D, and HOMA index	VLCKD (800 kcal/day, C: 28 g (14.6%), F: 35 g (38.7%), P: 85 g (46.7%), 2–2.5 L water/day + interval training 2 times a week 30–35 min/day via the Zoom platform	6 weeks after COVID-19	- reduction of body weight, BMI, amount of body fat, maintenance of lean mass, - improved muscle strength and physical performance - reduction in blood pressure, insulin, glycemia and HOMA, TG, LDL, and creatinine levels - increase in HDL	Camajani et al. [179]
3	A real-world, retrospective exploratory analysis in outpatients with type 2 diabetes, obesity, and positive COVID-19	n = 339, 54.9 ± 8.7 age	hemoglobin A1c, blood glucose, ketone bodies, body weight and body mass index, and assessment of the need for hospitalization	KD—C: <50 g, ~5–10%, P: 1.5 g/kg reference body weight, ~15–20%, F: to satiety ~70–75%	before and during treat-ment	greater weight loss was significantly associated with a lower likelihood of hospitalization	Volk et al. [180]

KD—ketogenic diet, ESD—eucaloric standard diet, EKD—eucaloric ketogenic diet, VLCKD—very low-energy ketogenic diet, P—protein, F—fat, C—carbohydrates, BMI—body mass index, TG—triglycerides, HOMA index—insulin resistance index, hemoglobin A1c—glycated hemoglobin.

## Data Availability

Data sharing is not applicable to this article.

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
