# Peer review of "Ketogenic Diet and Ketone Bodies as Clinical Support for the Treatment of SARS-CoV-2—Review of the Evidence"

_viruses, 2023, doi:10.3390/v15061262_

Round 1
Reviewer 1 Report
Comments and Suggestions for Authors
The manuscript “Ketogenic diet and ketone bodies as clinical support for the 3 treatment of SARS-CoV-2 - is the evidence sufficient?” is a very interesting paper in which authors summarized the evidence from tissue, animal, and human models and looked at the mechanisms of action of KD/ketone bodies against 13 COVID-19 in a very comprehensive way. The manuscript provides a large amount of data.
The manuscript is complete and the different sections of the article are well balanced and adequately supported by the data provided.
The manuscript is clear and answers the objectives. The manuscript is appropriate from the aims and scope of the journal and is potentially publishable "Viruses" with some minor changes.
1. Background need to be improved with more support from recent studies published in tire 4 journal particularly Science and Nature
2. The quality of Figure 3 is poor. Need to redraw the mechanism
3. Some more studies regarding efficacy of KD/ketone bodies in SARS-CoV-2 on biological material from patients with COVID-19 from developing countries need to be included in Table 3
4. Different study design can be compared to find the best one, if possible
5. Conclusion section need to improve. Must be based on the material provided in manuscript
6. The whole article need to be checked for typos and grammar errors
7. All reference should be according to journal guidelines
Author Response
Reviewer 1
The manuscript “Ketogenic diet and ketone bodies as clinical support for the 3 treatment of SARS-CoV-2 - is the evidence sufficient?” is a very interesting paper in which authors summarized the evidence from tissue, animal, and human models and looked at the mechanisms of action of KD/ketone bodies against 13 COVID-19 in a very comprehensive way. The manuscript provides a large amount of data.
The manuscript is complete and the different sections of the article are well balanced and adequately supported by the data provided.
The manuscript is clear and answers the objectives. The manuscript is appropriate from the aims and scope of the journal and is potentially publishable "Viruses" with some minor changes.
Response:
Thank you very much for taking the time to review the manuscript ''Ketogenic diet and ketone bodies as clinical support for the treatment of SARS-CoV-2 - review of the evidence'' and for your valuable comments and suggestions. We are very pleased that the topic of the manuscript aroused interest and thank you very much for your kind words. We have corrected the manuscript according to the recommendations, comments and suggestions as best we could. If we have misunderstood a point, we would like the opportunity to supplement our response.
Below are the responses to each point in the Review, with corrections to the text highlighted in colour in the text of the manuscript.
Reviewer
- Background need to be improved with more support from recent studies published in tire 4 journal particularly Science and Nature
Response: Thank you very much for your suggestion. We have corrected the background and included studies from recent years published in Science and Nature.
Reviewer
- The quality of Figure 3 is poor. Need to redraw the mechanism
Response: Thank you very much for drawing our attention to the quality of figure 3. We have corrected it as recommended. We think that in this form its quality will be satisfactory.
Reviewer
- Some more studies regarding efficacy of KD/ketone bodies in SARS-CoV-2 on biological material from patients with COVID-19 from developing countries need to be included in Table 3
Response: Thank you very much for your good point, that would be very much appreciated. However, we did not find such studies in the databases available to us. We only found a study from China in the form of a non-peer-reviewed preprint and included it. (However, we have a doubt here whether we can count China as a developing or developed country).
We also found draft studies including one retracted study - we have not included these. If we could ask you to possibly identify such studies it would be important for us to discuss the findings in the future and to complete this manuscript.
Reviewer
- Different study design can be compared to find the best one, if possible
Response: Thank you very much for this suggestion. Given the difficulty in comparing the studies we have presented here, we have included a note in the summary that it is important to note the limitations of the studies analysed, such as the small sample size and the lack of randomised controlled trials, which may affect the generalisability of the results.
Reviewer
- Conclusion section need to improve. Must be based on the material provided in manuscript
Response: Thank you very much for this comment. We have expanded the conclusions section to include sections from the manuscript. We have also added the benefits and risks of introducing nutritional therapy based on KD or ketone bodies and highlighted directions for future research. We hope that the quality of this section of the manuscript has improved significantly.
Reviewer
- The whole article need to be checked for typos and grammar errors
Response: Thank you very much for this suggestion. We have corrected the typos and grammatical errors that we were able to catch. However, if necessary, we will consult the manuscript with a native speaker of the language
Reviewer
- All reference should be according to journal guidelines
Response: Thank you very much for this comment. We have corrected the references in accordance with the journal's guidelines.
With best regards
On behalf of all the authors
Izabela Bolesławska

Reviewer 2 Report
Comments and Suggestions for Authors
For the potential use of a ketogenic diet and ketone bodies as clinical support for the treatment of SARS-CoV-2,the article provides a detailed account of the evidence from various models, including human, animal, and tissue studies. The authors have presented a strong argument for the use of β-hydroxybutyrate (BHB) to prevent metabolic reprogramming associated with COVID-19 infection, improve mitochondrial function, and reduce glycolysis in CD4+ lymphocytes. Moreover, the authors had also provided evidence that the use of KD/ketone bodies could provide an alternative carbon source for oxidative phosphorylation (OXPHOS) at the stage of virus entry into the host cell. The authors also reported that the use of KD and ketone bodies appeared to support the host immune response through multiple mechanisms. Animal models showed that KD resulted in protection against weight loss and hypoxemia, faster recovery, reduced lung injury, and better survival of young mice. Human studies indicated that KD increased survival, reduced the need for hospitalization for COVID-19, and showed a protective role against metabolic abnormalities after COVID-19. However, the authors should highlight the potential adverse effects of SARS-CoV-2 infection-induced ketoacidosis, which were mentioned in passing in the manuscript. Additionally, it is crucial to acknowledge the limitations of the studies reviewed, such as small sample sizes and lack of randomized controlled trials, which could influence the generalizability of the results. Overall, this manuscript provides a compelling case for the potential use of KD and ketone bodies as clinical nutritional interventions to assist in the treatment of COVID-19. However, further research is needed to validate these findings, and the authors should carefully consider the potential risks and benefits of this approach in the context of COVID-19 management.
Comments on the Quality of English LanguageMinor editing of English language required.
Author Response
Reviewer 2
For the potential use of a ketogenic diet and ketone bodies as clinical support for the treatment of SARS-CoV-2,the article provides a detailed account of the evidence from various models, including human, animal, and tissue studies. The authors have presented a strong argument for the use of β-hydroxybutyrate (BHB) to prevent metabolic reprogramming associated with COVID-19 infection, improve mitochondrial function, and reduce glycolysis in CD4+ lymphocytes. Moreover, the authors had also provided evidence that the use of KD/ketone bodies could provide an alternative carbon source for oxidative phosphorylation (OXPHOS) at the stage of virus entry into the host cell. The authors also reported that the use of KD and ketone bodies appeared to support the host immune response through multiple mechanisms. Animal models showed that KD resulted in protection against weight loss and hypoxemia, faster recovery, reduced lung injury, and better survival of young mice. Human studies indicated that KD increased survival, reduced the need for hospitalization for COVID-19, and showed a protective role against metabolic abnormalities after COVID-19. However, the authors should highlight the potential adverse effects of SARS-CoV-2 infection-induced ketoacidosis, which were mentioned in passing in the manuscript. Additionally, it is crucial to acknowledge the limitations of the studies reviewed, such as small sample sizes and lack of randomized controlled trials, which could influence the generalizability of the results. Overall, this manuscript provides a compelling case for the potential use of KD and ketone bodies as clinical nutritional interventions to assist in the treatment of COVID-19. However, further research is needed to validate these findings, and the authors should carefully consider the potential risks and benefits of this approach in the context of COVID-19 management.
Response:
Thank you very much for taking the time to review the manuscript ''Ketogenic diet and ketone bodies as clinical support for the treatment of SARS-CoV-2 - review of the evidence'' and for your valuable comments and suggestions. We are very pleased that the topic of the manuscript aroused interest and thank you very much for your kind words. We have corrected the manuscript according to the recommendations, comments and suggestions as best we could. If we have misunderstood a point, we would like the opportunity to supplement our response.
Below are the responses to each point in the Review, with corrections to the text highlighted in colour in the text of the manuscript.
Reviewer: However, the authors should highlight the potential adverse effects of SARS-CoV-2 infection-induced ketoacidosis, which were mentioned in passing in the manuscript.
Response: Thank you very much for this comment. We have expanded the introduction to include issues related to the effects of ketoacidosis in SARS-CoV-2 infection. We have also mentioned ketoacidosis in the conclusions as a possible risk to be considered when using KD in vulnerable patients (e.g. diabetes).
Reviewer: Additionally, it is crucial to acknowledge the limitations of the studies reviewed, such as small sample sizes and lack of randomized controlled trials, which could influence the generalizability of the results.
Response: Thank you very much for this suggestion. We have implemented it in the conclusions.
Reviewer: However, further research is needed to validate these findings, and the authors should carefully consider the potential risks and benefits of this approach in the context of COVID-19 management.
Response: Thank you very much for your valuable comment. We have included information about the need for further research to confirm the results presented in the conclusions. There, we also described the potential risks and benefits of such a nutritional approach in the context of COVID-19 management.
Reviewer: Comments on the Quality of English Language- Minor editing of English language required.
Response: Thank you very much for this suggestion. We have corrected the typos and grammatical errors that we were able to catch. However, we will consult the manuscript with a native speaker if necessary.
With best regards
On behalf of all authors
Izabela Bolesławska

Reviewer 3 Report
Comments and Suggestions for Authors
The manuscript presented by Izabela BolesÅ‚awska and et al. entitled “Ketogenic diet and ketone bodies as clinical support for the treatment of SARS-CoV-2 - is the evidence sufficient?” is of interest, well-written, clear, and easy to read. The topic is very interesting and, therefore, in the area of prevention, and treatment of SARS-CoV-2 with a ketogenic diet regimen, adding useful information on how to manage the infection even for future pandemics.
KD biochemical action is also depending on the epigenetic and antioxidant effect that diet induces. I suggest the authors include in the introduction section more details concerning this. PMID: 35956421, PMID: 32192146 and PMID: 31382449
Author Response
Reviewer 3
The manuscript presented by Izabela BolesÅ‚awska and et al. entitled “Ketogenic diet and ketone bodies as clinical support for the treatment of SARS-CoV-2 - is the evidence sufficient” is of interest, well-written, clear, and easy to read. The topic is very interesting and, therefore, in the area of prevention, and treatment of SARS-CoV-2 with a ketogenic diet regimen, adding useful information on how to manage the infection even for future pandemics.
KD biochemical action is also depending on the epigenetic and antioxidant effect that diet induces. I suggest the authors include in the introduction section more details concerning this. PMID: 35956421, PMID: 32192146 and PMID: 31382449
Response:
Thank you very much for taking the time to review the manuscript " Ketogenic diet and ketone bodies as clinical support for the treatment of SARS-CoV-2 - review of the evidence" and for your valuable comments and suggestions. We are very pleased that the topic of the manuscript aroused interest and thank you very much for your kind words. We have taken into account the reviewer's very fair suggestion and included more details of the epigenetic and antioxidant effects caused by KD and ketone bodies in the introductory section using the literature items indicated. We have revised the manuscript as recommended to the best of our ability. The corrections have been highlighted in colour in the text of the manuscript.
With best regards
On behalf of all authors
Izabela Bolesławska

Reviewer 4 Report
Comments and Suggestions for Authors
The article entitled as "Ketogenic diet and ketone bodies as clinical support for the treatment of SARS-CoV-2 is the evidence sufficient?" is no doubt interesting compilation. However, I have several suggestions to be addressed before the publication.
Firstly, the title, I personally believe the title is loose in context of the scientific terms. Try to modify the title which will be more suitable.
SARS-CoV-2 epidemic? Please Check is it pandemic or epidemic?
Provide the abbreviation first in the intro. at line no. 29, SARS-CoV-2 (Severe Acute Respiratory Syndrome Coronavirus 2)
Improve the introduction, while highlighting the relevance of your review of literature.
Why are such therapeutic interventions still possessing relevance amid the pandemic is being halt by vaccinations?
As a reader, I would like to know more about the relevance and importance.
Figure 1: The quality of figure is not up to the mark. There are several well elaborated figures represents the entry and penetration of SARS-CoV-2 in the host cell. Try to improve the figure.
If possible, please check the following article:
https://doi.org/10.3390/medicina59030507 for Figure 1.
Figure 3: Some of the metabolites names are not clear.
Please check for the quality purposes.
Figure 4: Check the spellings of angiotensin inside the figure.
Section 4.2: Effectiveness of KD/ketone bodies in regulating innate and acquired immunity
It will be interesting to see this section in the form of a figure that how KD/KB can regulate the immune response?
And you can show such regulation in the development of severe form of the disease or the asymptomatic disease. How can such imbalance lead to any form of the severity?
It will be an interesting figure.
I found conclusion has been written very poorly. Please improve the conclusion holistically and try to provide all of your major highlights.
Additionally, provide the future directions that how this work will be used in the future etc.
Best Wishes
Comments on the Quality of English Language
English language used is acceptable. However, final checks are required to avoid any potential grammatical errors.
Author Response
Reviewer 4
Comments and Suggestions for Authors
The article entitled as "Ketogenic diet and ketone bodies as clinical support for the treatment of SARS-CoV-2 is the evidence sufficient?" is no doubt interesting compilation. However, I have several suggestions to be addressed before the publication.
Thank you very much for taking the time to review the manuscript 'Ketogenic diet and ketone bodies as clinical support for the treatment of SARS-CoV-2 - review of the evidence' and for your valuable comments and suggestions. We are very pleased that the topic of the manuscript aroused interest and thank you very much for your kind words. We have taken into account all the reviewer's comments and suggestions. We have corrected the manuscript as recommended to the best of our ability. The corrections have been highlighted in colour in the text of the manuscript.
Detailed responses:
Reviewer: Firstly, the title, I personally believe the title is loose in context of the scientific terms. Try to modify the title which will be more suitable.
Response: Thank you very much for your suggestion. We have changed the title, with the hope that it will be more appropriate.
Reviewer: SARS-CoV-2 epidemic? Please Check is it pandemic or epidemic?
Response: Thank you very much for your attention. Of course, the SARS-CoV-2 outbreak on March 11, 2020 has been declared a pandemic by the World Health Organization. We have corrected our mistake.
Reviewer: Provide the abbreviation first in the intro. at line no. 29, SARS-CoV-2 (Severe Acute Respiratory Syndrome Coronavirus 2)
Response: Thank you very much for your comment. We have added an explanation of the abbreviation the first time it is used.
Reviewer: Improve the introduction, while highlighting the relevance of your review of literature.
Response: Thank you very much for your suggestion. We have corrected the introduction taking into account the relevance of the literature review done.
Reviewer: Why are such therapeutic interventions still possessing relevance amid the pandemic is being halt by vaccinations? As a reader, I would like to know more about the relevance and importance.
Response: Thank you very much for your comment. We have complied and explained in the introduction the need for therapeutic interventions despite the SARS-CoV-2 pandemic being contained by vaccines.
Reviewer: Figure 1: The quality of figure is not up to the mark. There are several well elaborated figures represents the entry and penetration of SARS-CoV-2 in the host cell. Try to improve the figure.
If possible, please check the following article:
https://doi.org/10.3390/medicina59030507 for Figure 1.
Response: We have indeed presented the penetration of the SARS-CoV-2 virus in an over-simplified way and thank you very much for drawing attention to the well-developed drawings. This is very important to us. Our initial intention was to present the penetration of the virus in as schematic a form as possible in order to balance the chapters. We currently feel that it is better to remove the drawing from the text - which we have done.
Reviewer: Figure 3: Some of the metabolites names are not clear.
Please check for the quality purposes.
Response: Thank you very much for your comment. This has been corrected.
Reviewer: Figure 4: Check the spellings of angiotensin inside the figure.
Response: Thank you very much for your comment. This has been corrected.
Reviewer: Section 4.2: Effectiveness of KD/ketone bodies in regulating innate and acquired immunity
It will be interesting to see this section in the form of a figure that how KD/KB can regulate the immune response? And you can show such regulation in the development of severe form of the disease or the asymptomatic disease. How can such imbalance lead to any form of the severity? It will be an interesting figure.
Response: thank you very much, this is indeed a very interesting suggestion. We have presented the support of immune regulation by KD/ketone bodies in SARS-CoV-2 in the form of a drawing, however we are not sure if this is what the reviewer had in mind. If this drawing is too simplistic we kindly ask you to allow us to improve it.
Reviewer: I found conclusion has been written very poorly. Please improve the conclusion holistically and try to provide all of your major highlights. Additionally, provide the future directions that how this work will be used in the future etc.
Response: Thank you very much for these comments. We have expanded the conclusions section to include information from the manuscript. We have also added the benefits and risks of introducing nutritional therapy based on KD or ketone bodies and highlighted directions for future research. We hope that the quality of this section of the manuscript has significantly improved.
Best Wishes
Comments on the Quality of English Language
English language used is acceptable. However, final checks are required to avoid any potential grammatical errors.
Response: Thank you very much for this suggestion. We have corrected the typos and grammatical errors that we were able to catch. However, we will consult the manuscript with a native speaker if necessary.
With best regards
On behalf of all the authors
Izabela Bolesławska

Round 2
Reviewer 4 Report
Comments and Suggestions for Authors
The authors have revised the manuscript as per the suggestions. However, I suggest the following to consider carefully.
I suggest not to remove the figure 1, instead modify it and incorporate it. As it will give the understanding of infection mechanism.
Secondly the figure legends,
Like course of COVID-19, Explain the figure legends like high impact journal articles.
This will make readers to understand the relevance of the figures
My major concern is the conclusion, Rather than stretching the conclusion with references. Try to write in a crisp way which will give the major highlights of the manuscript.
This conclusion section doesn't fit with the manuscript.
I suggest please look into the conclusion and write the future directions section separately.
Best Wishes
Comments on the Quality of English Language
English language is acceptable
Author Response
Reviewer 4
Reviewer
The authors have revised the manuscript as per the suggestions. However, I suggest the following to consider carefully.
Response: Thank you very much for allowing us to revise the manuscript "Ketogenic diet and ketone bodies as clinical support for the treatment of SARS-CoV-2 - review of the evidence" and for your very pertinent comments.
We have addressed the reviewer's suggestions. We have made the necessary corrections. However, if we have misunderstood a subsection, please allow us to correct it.
We have marked our changes in colour in the text of the manuscript.
Reviewer
I suggest not to remove the figure 1, instead modify it and incorporate it. As it will give the understanding of infection mechanism.
Response: Thank you very much for drawing attention to this issue. Indeed, the introduction of Figure 1 provides a better understanding of the mechanism of entry of the SARS-CoV-2 virus.
Reviewer
Secondly the figure legends,
Like course of COVID-19, Explain the figure legends like high impact journal articles.
This will make readers to understand the relevance of the figures
Response: Thank you very much for this comment. We have corrected the legends and added descriptions. It does indeed add to their readability.
Reviewer
My major concern is the conclusion, Rather than stretching the conclusion with references. Try to write in a crisp way which will give the major highlights of the manuscript.
This conclusion section doesn't fit with the manuscript.
I suggest please look into the conclusion and write the future directions section separately.
Response: Thank you very much for the reviewer's suggestion. We have added a new subsection 4.6 Future research directions and there we have included the text that really stretched the conclusion unnecessarily. The conclusion is now more concrete and concise.
Best Wishes
Thank you again for your tremendous help.
Yours sincerely
On behalf of all authors
Izabela Bolesławska
